# Translocation of Oral Microbiota into the Pancreatic Ductal Adenocarcinoma Tumor Microenvironment

**DOI:** 10.3390/microorganisms11061466

**Published:** 2023-05-31

**Authors:** Kelley N. L. McKinley, Kelly M. Herremans, Andrea N. Riner, Vignesh Vudatha, Devon C. Freudenberger, Steven J. Hughes, Eric W. Triplett, Jose G. Trevino

**Affiliations:** 1Department of Microbiology and Cell Science, University of Florida, Gainesville, FL 32611, USA; pposse@ufl.edu; 2Department of Surgery, College of Medicine, University of Florida, Gainesville, FL 32610, USA; kelly.herremans@surgery.ufl.edu (K.M.H.); andrea.riner@surgery.ufl.edu (A.N.R.); steven.hughes@surgery.ufl.edu (S.J.H.); 3Division of Surgical Oncology, Virginia Commonwealth University, Richmond, VA 23298, USA; vignesh.vudatha@vcuhealth.org (V.V.); devon.freudenberger@vcuhealth.org (D.C.F.); 4Massey Cancer Center, Virginia Commonwealth University, Richmond, VA 23298, USA

**Keywords:** pancreas, tumor, saliva, microbiome, *Veillonella*

## Abstract

Oral dysbiosis has long been associated with pancreatic ductal adenocarcinoma (PDAC). In this work, we explore the relationship between the oral and tumor microbiomes of patients diagnosed with PDAC. Salivary and tumor microbiomes were analyzed using a variety of sequencing methods, resulting in a high prevalence and relative abundance of oral bacteria, particularly *Veillonella* and *Streptococcus*, within tumor tissue. The most prevalent and abundant taxon found within both saliva and tumor tissue samples, *Veillonella atypica*, was cultured from patient saliva, sequenced and annotated, identifying genes that potentially contribute to tumorigenesis. High sequence similarity was observed between sequences recovered from patient matched saliva and tumor tissue, indicating that the taxa found in PDAC tumors may derive from the mouth. These findings may have clinical implications in the care and treatment of patients diagnosed with PDAC.

## 1. Introduction

Pancreatic cancer remains a deadly disease and it is poised to become the second leading cause of cancer-related death within the next decade [1]. This is in part due to limited diagnostic and treatment modalities. Poor oral health and edentulism have long been associated with the development of pancreatic cancer [2]. Large-scale prospective studies and meta-analyses have shown that periodontitis, tooth loss and oral lesions are associated with up to an 80% excess risk of developing pancreatic cancer, even when accounting for confounding factors such as smoking [3]. These well-powered studies set the groundwork for additional investigation into the underlying cause of this broad association between oral health and pancreatic cancer.

Dysbiosis of the oral microbiome has also been associated with the development of pancreatic ductal adenocarcinoma (PDAC). To assess the oral microbiome of patients who developed PDAC, Fan et al., first performed a nested case–control utilizing two large prospective cohorts, the American Cancer Society Cancer Prevention Study II and the National Cancer Institute Prostate, Lung, Colorectal and Ovarian Cancer Screening Trial [4]. They found that patients who eventually developed PDAC had increased abundance of *Porphyromonas gingivalis* and *Aggregatibacter actinomycetemcomitans* within their oral microbiome. These changes were associated with nearly a twofold risk of developing PDAC and could be detected up to 10 years prior to their cancer diagnosis [4].

Furthermore, bacteria commonly found in the oral cavity, including *Megasphaera*, *Veillonella*, *Streptococcus*, and *Fusobacterium* have been identified in PDAC tumor tissue [5,6,7,8]. According to a study by Huang et al., an increased relative abundance of *Megasphaera*, among others, within PDAC tissue was associated with long-term survival. Administration of *Megasphera* was further shown to improve tumor growth administration when combined with anti-programmed cell death-1 (anti-PD-1) treatment in a murine model. Conversely, *Porphyromonas* and *Aggregatibacter* within tumor tissue were associated with short-term survival in PDAC [7].

However, little remains known about how oral bacteria may impact the pancreas or specifically how the tumor microbiome is established. It has previously been proposed that bacteria in the pancreas translocate from the gut; however, translocation from the mouth and bloodstream have also been proposed [9,10,11]. Translocation from the mouth to distant sites has previously been implicated in other disease states including atherosclerosis, adverse pregnancy outcomes, rheumatoid arthritis, organ abscesses and endocarditis [12].

Microbiome association studies continue to suggest that oral bacteria play a role within the PDAC tumor microenvironment; however, that role is has yet to be defined and current research aims to identify key taxa of interest. Associations between the oral microbiome and PDAC have been proposed to improve diagnostic and treatment modalities. However, data are limited on how the oral microbiome may influence the pancreatic microbiome and ultimately pancreatic tumor biology. The aim of this study was to assess the presence of oral bacteria within the pancreatic tumor microbiome while further exploring the potential mechanisms of tumorigenesis of the most abundant and prevalent oral bacteria found within the tumor microbiome.

## 2. Materials and Methods

### 2.1. Experimental Design and Cohort

The initial design for this exploratory project was to increase understanding of the bacterial composition of saliva and pancreatic tumors of patients diagnosed with PDAC. Data showing the presence of certain bacteria within tumors highlighted a potential relationship with the saliva microbiome. To explore the possibility of oral translocation of bacteria into the tumor tissue, patient-matched saliva and tumor samples were further investigated with expanded sequencing for greater taxonomic resolution at the species level. The most abundant and prevalent species of oral bacteria found within both saliva and tumor tissue samples were cultured for full genomic resolution to investigate potential mechanisms of pathogenesis.

There were 12 women and 17 men, totaling 29 PDAC patients, from the North Florida region enrolled in this study. The cohort had a median age of 67 years old. The racial makeup of this cohort was predominately non-Hispanic white, accounting for 90% of patients enrolled. Previous history of diabetes was present in 40% of the cohort. This work was approved by the University of Florida Institutional Review Board and assigned approval number IRB20160087.

### 2.2. Sample Collection and Storage

Saliva and tumor samples were collected from patients diagnosed with PDAC. Saliva samples were collected using the passive drool technique into a sterile saliva collection container and immediately aliquoted and stored at −80 °C until further processing. Pancreatic tumor samples were taken directly from the operating room in sterile conditions and immediately frozen and stored at −80 °C until further processing.

### 2.3. DNA Isolation and 16S rRNA Sequencing of Saliva Samples

DNA was extracted from 100 uL of each saliva sample using the Isohelix Xtreme DNA Isolation Kit following manufacturer’s protocol (Cell Projects Ltd., Kent, UK). Purification of isolated DNA was carried out using the Zymo Genomic DNA Clean & Concentrator Kit (Zymo Research, Irvine, CA, USA). The 16S rRNA v3-v4 PCR and sequencing was carried out as previously described [13]. Sequence reads were processed into amplicon sequencing variants (ASVs) as previously detailed [14].

### 2.4. DNA Isolation and 16S rRNA Sequencing of Tumor Tissue Samples

DNA was extracted from ~35 mg of each tumor tissue sample using the Qiagen QiAamp Fast DNA Tissue Kit using the manufacturer’s protocol. Purification of isolated DNA was carried out using the Zymo Genomic DNA Clean & Concentrator Kit. Nested PCR was used for tumor samples using full length 16S primers 8F and 1492R with initial denaturation of 95 °C for 2 min, 20 cycles of 95 °C for 30 s, 58 °C for 30 s, and 72 °C for 90 s, with a final step of elongation at 75 °C for 5 min. The full length 16S rRNA amplicons were subjected to a second round of PCR using Illumina barcoded 341F and 806R primers targeting the v3-v4 region using the same thermocycler parameters used for the saliva PCR. The 16S rRNA v3-v4 amplicons were purified by gel extraction using the E.Z.N.A. Gel Extraction Kit following the manufacturer’s protocol (Omega Bio-tek, Doraville, CA, USA). The 16S rRNA v3-v4 sequencing followed the same protocol as the saliva samples. Sequence reads were processed into amplicon sequencing variants (ASVs) as previously detailed [14].

### 2.5. Rrn Operon Sequencing

Rrn operon sequencing was carried out on a subset of saliva and tumor samples as previously described using an Oxford Nanopore Technology (ONT) R9.4 flow cell [15]. The described pipeline was repeated with different quality filters of quality score (QS) 15,16,17,18,19 and 20.

### 2.6. Bacterial Quantification by qPCR

Total bacterial quantification was carried out on a subset of saliva and tumor samples using universal 16S rRNA 341F and 806R primers as previously described [16]. Copies of 16S rRNA per g of tumor tissue or uL of saliva were calculated by multiplying the average copy number per reaction replicate by DNA isolation volume (50 uL) and dividing by the mass or volume of sample used for isolation.

### 2.7. Culture and Sequencing of Veillonella atypica

The selection of target genera for culture involved using the most abundant and prevalent taxa found within both 16S and rrn datasets to increase the likelihood of live cell recovery from available samples. Saliva and tumor samples from a patient with a high abundance of *Veillonella* ASVs were placed in a 24 h lactate enrichment broth of 5 g trypticase, 3 g yeast, 0.750 g sodium thioglycolate, 1.1 mL tween 80-, and 21 mL sodium lactate per liter of distilled H_2_O. Vancomycin was added to a final concentration of 7.5 mcg/mL and pH was stabilized at 7.5. The broth enrichment was serially diluted and plated on the same media with agar added. Cultures were grown anaerobically for 36 h and isolated. Culture stocks were stored in 50% glycerol at −80 °C.

DNA extraction of culture isolates was carried out using Qiagen Genomic-tips protocol as described in the Qiagen Genomic DNA handbook. Extracted DNA was sequenced with a Nanopore GridIon, using an R10.4 flow cell with accompanying Ligation Sequencing Kit Q20+ (SQK-LSK112) as described by the manufacturer (Oxford Nanopore Technologies, Oxford, UK). Sequenced reads were assembled using Flye 2.9 [17], where a single 2,110,364 bp contig was produced. This contig was polished using four rounds of Racon [18] and one round of Medaka 1.6.0 [19] before submission to the NCBI Prokaryotic Genome Annotation Pipeline [20]. The annotated genome was used for analysis in KEGG [21].

### 2.8. PCR-Free Sequencing of Tumor Tissue

Extracted and purified DNA from a tumor tissue sample of interest was sequenced using Illumina NovaSeq 6000 SP 2 × 150 (ICBR, Gainesville, FL, USA). Sequenced reads were joined and filtered using fastq-join [22] and fastp [23], respectively. Human host read filtering and bacterial read mapping were carried out using minimap2 [24].

## 3. Results

### 3.1. Microbiome Profiles by Sample Type and Patient-Matched Samples Using 16S rRNA

Initial bacterial profiling was carried out using standard 16S rRNA v3-v4 PCR and Illumina MiSeq 2 × 300 sequencing. Oral microbiome profiles exhibited typical salivary profiles, with *Veillonella*, *Streptococcus*, *Megasphaera*, *Prevotella*, *Actinomyces*, *TM7x*, *Leptotrichia*, *Oribacterium*, *Lachnoanaerobaculum*, and *Atopobium* in 90% of samples (Appendix A). *Veillonella* and *Streptococcus* were found in the oral microbiome of all patients (Figure 1). *Veillonella* had the most relative abundance within all but two saliva samples, followed by *Streptococcus* and *Megasphaera*.

Taxa found in pancreatic tumors varied widely between patients as compared to saliva (Appendix A). However, *Veillonella*, *Streptococcus*, *Lactobacillus*, *Klebsiella*, *Prevotella*, *Stentrophomonas*, and *Devosia* were present among at least 35% of tumors. The most prevalent genera within saliva were also found in the pancreatic tumor microbiome, with *Veillonella* and *Streptococcus* present in 92% and 64% of samples, respectively (Figure 2). Relative abundance of *Veillonella* varied by patient but it was the most abundant genera in four tumor samples.

To investigate the origin of taxa found within the tumor microbiome, patient tumor tissue and saliva samples were matched, resulting in a 7-patient subset. *Veillonella*, *Streptococcus*, and *Megasphaera* were found in nearly all matched sample pairs (Figure 3), with *Veillonella* present in all matched sample pairs.

### 3.2. RRN Sequencing of PDAC Subset Highlights Specific Taxa Similarity

To further confirm the similarities of the oral and pancreatic tumor microbiomes using an alternative method, rrn sequencing was performed on a subset of 5 patients with matched saliva and pancreatic tumor samples. The rrn operon, including the 16S-ITS-23S regions, was chosen as the target sequence for analysis because the added length of the rrn sequence, as compared to the 16S rRNA v3-v4 region, gives greater taxonomic resolution at the species level. Results were similar to the 16S rRNA analysis, with *Veillonella* and *Streptococcus* being the dominant genera (Appendix A). At the species level, taxa and relative abundance exhibited more variation between patient and sample types. The most prevalent species were *Veillonella atypica*, *Veillonella dispar*, and *Streptococcus mitis*, appearing in 60% of total patient samples, with *Veillonella atypica* being present in 60% of the sample pairs (Figure 4).

Some of the rrn operon results differed from previous 16S rRNA results, with one of the pancreatic tumors recovering no sequences and one of the saliva samples recovering sequences from only one species. A quality score cutoff of 15 was used for this rrn analysis as it generated the best results, balancing quality with taxa composition. However, after reducing the quality score cutoff to 10, both the saliva and tumors show several different taxa.

### 3.3. Culture and Sequencing of Patient-Derived Veillonella atypica

To investigate the genomic similarities of *Veillonella* from the oral and pancreatic tumor microbiomes, culture and long read sequencing was performed. *Veillonella atypica* was selected for culture due to it having greater prevalence and abundance within both oral and pancreatic tumor microbiomes than any other species. Saliva and pancreatic tumor samples from a single patient were used for culture and treated to the same growth conditions. Saliva culture was successful in producing isolates of *Veillonella atypica*, though culture of pancreatic tumors could not be successfully performed. However, previous studies have successfully cultured from PDAC tumors, showing that live bacteria do exist within them [25,26].

The difficulty in culturing from tumors was likely the result of the low quantity of bacteria generally found in tissue. Quantitative PCR results showed that the bacterial load of saliva and pancreatic tumor sample from this patient were 1.2 × 105, and 104 copy numbers per reaction, respectively. The qPCR results for the remaining of the rrn subset of patients follows a similar ratio of 1000:1 saliva to pancreatic tumor tissue for copy numbers per mL of isolated DNA.

Direct ONT sequencing of the *V. atypica* isolate cultured from saliva produced 2.3 trillion bases, totaling 1.2 million reads, with a mean read length of 1.8 k bp and mean quality score of 16.1. These reads were assembled into a complete 2.1 million base pair long circular genome with 1049-fold coverage.

### 3.4. Deep Sequencing of Pancreatic Tumor Microbiome

In order to obtain further evidence for *V. atypica* within the tumor, deep sequencing of pancreatic tumor DNA was performed using the NovaSeq 6000 platform, resulting in 2.4 billion paired end reads of 2 × 150 bp in length for a total of 363 billion bases. As expected, the vast majority of those reads were human. After host-filtering, and mapping to the genome of *V. atypica* cultured from patient-matched saliva, only 1098 reads remained, demonstrating the low bacterial load within pancreatic tissue. Of the reads that mapped to the *V. atypica* culture genome, 15 had greater than 90% sequence similarity, using NCBI Blast.

The probability that any 150 bp read among the 2.4 billion reads would be an identical match to the *V. atypica* genome is 1 in 2.0 × 1090. Hence, it is extremely unlikely that an exact match for a 150 bp read can be obtained by chance in this dataset. The amount of human DNA sequenced from the tumor sample was the equivalent of the amount of DNA in 60.5 human cells. The amount of *V. atypica* DNA identified in the DNA sequenced was equivalent to the amount of DNA in 0.0045 *V. atypica* cells. Hence, in the sample sequenced, the ratio of *V. atypica* cells to human cells was 1:13,333.

The probabilities above are persuasive that *V. atypica* cells were present in the tumor. However, it is not known whether these bacterial cells were alive in the tumor at the time of tumor isolation because culturing these bacteria from this small tumor sample failed. This culturing failure may be caused by the possibility that the bacterial cells were not alive in the tumor. However, as the calculation in the above paragraph shows, the *V. atypica* cells were quite rare in this tumor sample. As a result, the very small tumor samples used may have been insufficient for culturing. It may also be that the frozen storage conditions used for the tumor tissue did not allow long-term bacterial survival. Nevertheless, we can confidently say that *V. atypica* cells were within or on the tumor sample sequenced and that they were rare in number compared to the number of human cells.

To confirm the probabilities described above, the 363 billion bases were mapped against the genomes of other bacterial species that would not be expected in human tissues or might be present in the tumor based on 16S rRNA sequencing. As controls for this analysis, none of the 4.8 billion, 150 bp reads from tumor sequencing matched bacterial genomes associated with plants, *Bradyrhizobium japonicum* and *Liberbacter crescens*, that we would not expect to be associated with humans. Similarly, none of the reads precisely matched the genomes of two bacterial species found in the subject’s saliva, *Streptococcus mitis* and *Porphyromonas gingivalis*, bacteria genome of a bacterium found only in the saliva and not in the tumor. However, one read was found to match the genome precisely of a bacterium found in both saliva and tumor of a single subject.

### 3.5. V. atypica Genome Annotation and Functional Prediction Highlight Tumorigenesis Potential

The National Center for Biotechnology Information (NCBI) annotation for the assembled genome from the patient-derived *V. atypica* isolate showed 2031 total genes with 1966 coding sequences (CDS). Several genes of interest relating to potential tumorigenesis were mapped to our *V. atypica* genome. Utilizing the Kyoto Enclyclopedia of Genes and Genomes (KEGG), genes were present in pathways that may contribute to colonization of tumor tissue, co-aggregation with other bacteria and host immune response including nitrogen metabolism, quorum sensing, biofilm formation and lipopolysaccharide (LPS) biosynthesis (Table 1).

### 3.6. Veillonella atypica Sequence Similarity between Sample Types

*Veillonella atypica* sequences in patient-matched saliva and tumor tissue samples were found within both the 16S rRNA and rrn operon datasets. Two *V. atypica* ASVs shown in the 16S rRNA results were shared between saliva and tumors in 58% of patients with matched samples. With a baseline ONT quality score minimum of Q15, the rrn operon dataset showed *V. atypica* shared among sample types in 75% of the PDAC patients with matched saliva and tumor samples. To adequately compare sequence similarity, the quality score minimum was increased to Q18 and Q20, resulting in sequence similarities of 97% in one patient and 99% in another patient, respectively.

The assembled *V. atypica* genome cultured from patient saliva was used as a reference genome for sequence alignment of the rrn operon and NovaSeq data recovered from patient-matched pancreatic tumors. With a quality score minimum of Q20, there was a 99% sequence similarity between the assembled genome and the rrn operon. A single sequence from the NovaSeq data showed 100% sequence similarity with the assembled genome, indicating that the oral cavity is the likely origin of *V. atypica* found within the tumor tissue microbiome.

## 4. Discussion

The results of this study further support the relationship between the oral microbiome and pancreatic tumor microbiome in patients with PDAC. The bacterial profile within the oral microbiome of patients with PDAC exhibited a reasonably normal saliva profile, but a greater diversity among the tissue samples. However, throughout this diversity, oral bacteria, particularly *Veillonella* and *Streptococcus*, continued to be the dominant bacteria in most tissue samples. While *Veillonella* is a common oral bacterium, it is present in lower abundance within a healthy population compared to our findings, suggesting that oral dysbiosis may be occurring in patients with PDAC [30].

An abundance of oral bacteria, particularly *Veillonella* species, in PDAC tumor tissue has also been previously described [6,7,8,31]. Such a high prevalence and abundance of oral bacteria in PDAC tumors raises the question of how and why they are present. The prevalence and abundance of oral bacteria in PDAC tumors may be due to *Veillonella*’s role as a bridge species. It has been shown to be an early oral biofilm colonizer and co-aggregates with *Streptococcus* species [32,33]. Our findings support this notion, as *Streptococcus* was found to be the second most prevalent and abundant oral bacteria present in PDAC tumors. *Veillonella*, Gram-negative anaerobic cocci, are well known lactate fermenters and lactate may be an important contributing factor as to why these oral bacteria colonize tumor tissue. Pancreatic cancer cells have been shown to produce an excess amount of lactate, known as the Warburg effect, creating an ideal location for *Veillonella* to find a source of carbon and energy [34]. Furthermore, in a murine model, *Veillonella* have been shown to translocate systemically and colonize tumor tissue [26].

*Veillonella* have also been implicated in osteomyelitis, meningitis and endocarditis [35,36,37]. Evidence suggests that infective endocarditis and coronary artery disease may be initiated by the translocation of oral bacteria from an unhealthy mouth though the bloodstream, forming plaques within the heart [38]. Given the distal location of these sites compared to normal *Veillonella* colonization of the mouth, gut and vaginal cavity, translocation through systemic circulation is certainly plausible.

Poor oral health due to periodontal disease has also been associated with pancreatic cancer [2], lending to the possibility of translocation of oral bacteria through the gumline, into the bloodstream and colonizing PDAC tumors. While *Veillonella* has been shown to be present in the gut, it has a much lower abundance than what is seen in the oral cavity [39]. The existing associations between pancreatic cancer and oral health, the colonization of oral bacteria to distal sites, the low abundance of *Veillonella* in normal gut microbiomes as well as the sequence similarities identified in our datasets suggest that the pancreatic microbiome may arise not only from the gut, but from the oral microbiome.

This study further highlights potential interactions between the microbiota and pancreatic tumor. The patient derived *V. atypica* isolate genome contains genes involved with LPS biosynthesis and biofilm formation that could contribute to cancer progression or severity [40,41,42]. LPS, which may lead to both acute and chronic inflammation, has been implicated in the promotion of metastasis in therapeutic resistance, tumor growth and immunosuppression [40]. Genes for biofilm formation and quorum sensing were also found within this genome, highlighting the ability of *V. atypica* to support other bacterial types. Furthermore, biofilms have been previously described to confer treatment resistance and may even aid in harboring tumorigenic bacteria [43,44]. *Veillonella* have also been shown to alter tumor microenvironments in cervical and lung cancer [45,46,47]. In cervical cancer, *Veillonella* species have been shown to induce upregulation of pro-inflammatory mediators. In lung cancer, the presence of *Veillonella* within the lower airways is associated with disease progression. Further, in a murine model of lung cancer, *Veillonella* led to decreased survival, increased tumor burden, increased inflammatory mediators and activation of checkpoint inhibition. In contrast, researchers have also suggested *Veillonella* to have the ability to arrest tumor growth in mice for a short period of time [26]. While the limited information available is conflicting and no direct causation has been attributed to *Veillonella*, they certainly have the potential to play a role in cancer development.

This study provides novel insight into commensal bacteria of interest, *Veillonella*, that normally appear in the human mouth, gut and vaginal cavity. We showed a high prevalence and abundance of *V. atypica* across all patients and sample types. Comparing *V. atypica* sequences from patient-matched saliva and tumors showed very high sequence similarities, indicating possible direct oral translocation through the bloodstream. However, there were some limitations in this study that should be addressed. First, the sample size was limited in the matching cohort as only seven patients with both saliva and pancreatic tumor tissue were included. Second, patients in this cohort included only those with resectable, early stage PDAC disease, which may limit generalizability to patients with advanced metastatic disease. Third, patient stool samples were not collected, and while *V. atypica* sequence similarities between saliva and tumors were high, stool or other gastrointestinal samples would make source tracking via sequence similarity analysis more robust. Future studies should examine a greater number of patients with matching sample types including saliva, pancreatic tumors and stool. Fourth, the amount of pancreatic tumor tissue in each sample was small. The average weight of tissue samples were only 0.074 g, making culture very difficult and necessitating larger aliquots of tissue in future culture experiments. However, despite these limitations, this study elucidates the composition of the oral and pancreatic tumor microbiomes in patients diagnosed with PDAC. It further establishes a relationship between these two sites, specifically identifying the presence of oral *V. atypica* and exploring its translocation to PDAC tumors and its role in tumorigenesis. As we transition to the application of precision medicine in cancer treatment, further studies are necessary to assess the impact of the oral microbiota on PDAC tumor biology. These findings may lead to a potential target to ultimately improve patient outcomes in this deadly disease.

## Figures and Tables

**Figure 1 microorganisms-11-01466-f001:**
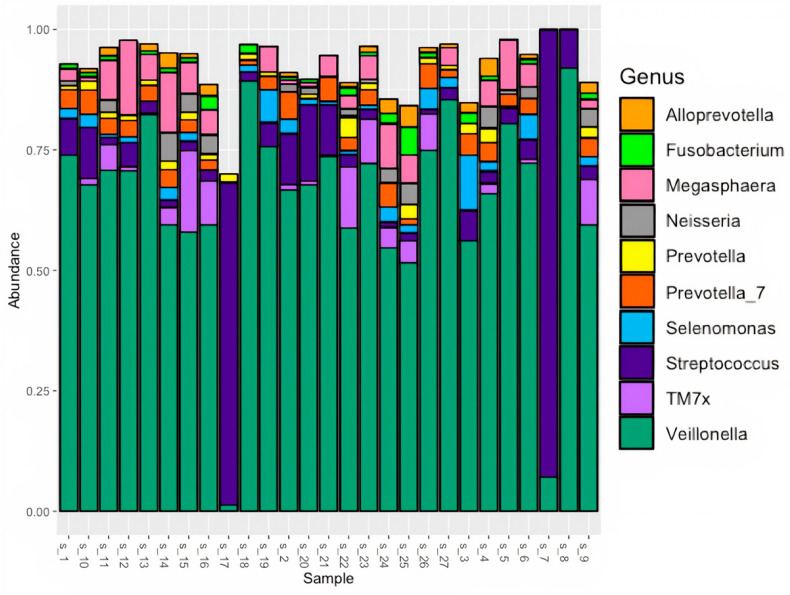
Relative abundance of top 10 genera in PDAC patient saliva using standard 16S rRNA v3-v4 sequencing.

**Figure 2 microorganisms-11-01466-f002:**
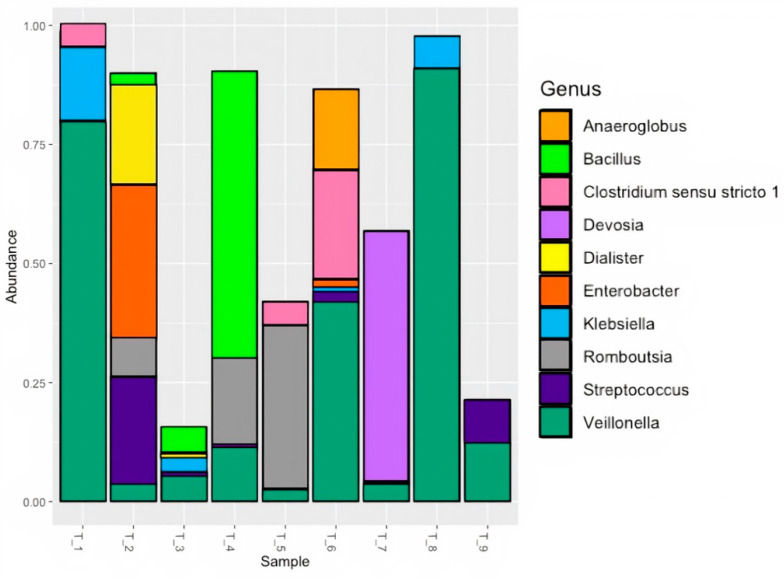
Relative abundance of top 10 genera in PDAC patient tumor tissue using standard 16S rRNA v3-v4 sequencing.

**Figure 3 microorganisms-11-01466-f003:**
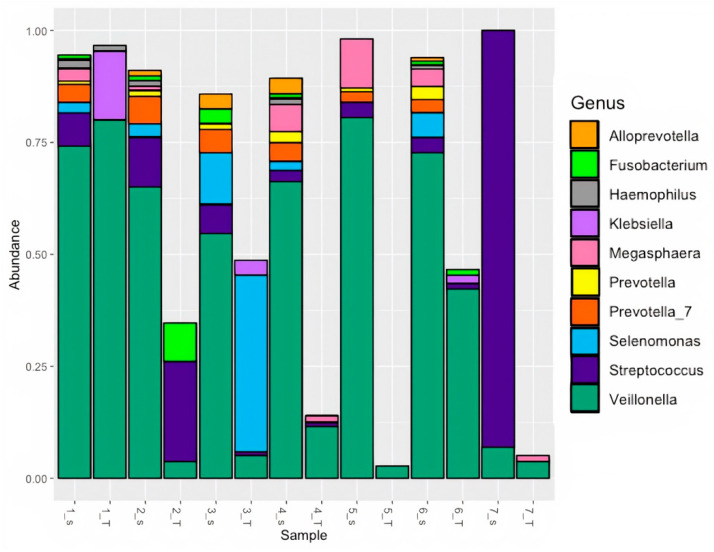
Relative abundance of top 10 genera in PDAC patient-matched saliva and tumors using standard 16S rRNA v3-v4 sequencing.

**Figure 4 microorganisms-11-01466-f004:**
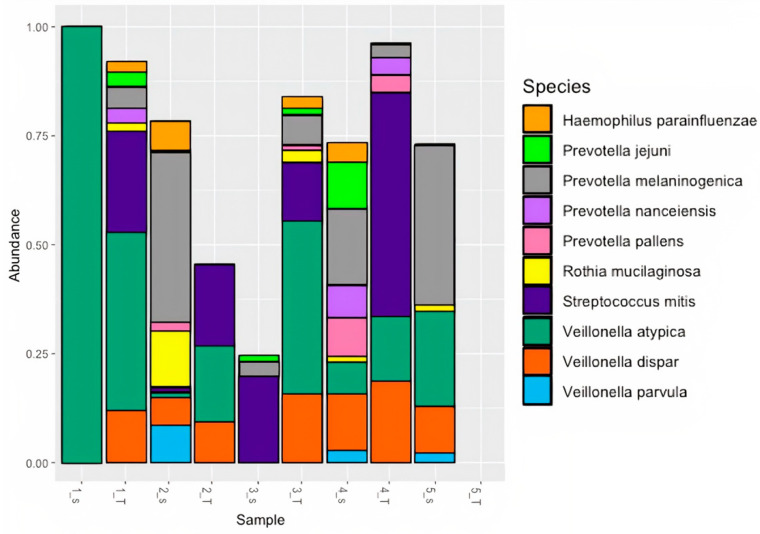
Relative abundance of top 10 species for PDAC subset using RRN sequencing.

**Table 1 microorganisms-11-01466-t001:** Genes of interest in *Veillonella atypica* culture isolate genome using KEGG.

Pathway	Genes of Interest	Potential Tumorigenesis	Reference
LPS Biosynthesis	*lpxA*, *lpxB*, *lpxC*, *lpxD*, *kdsA*, *kdsB*, *gmhA*, *rfaE1*, *rfaD*	Leads to expression of oncogenic pathways through TLR4 activation.	Massoumi et al. [27]
Nitrogen Metabolism	*narH*, *narI*, *hcp*, *gltB*	Inflammation-associated nitrate increases metabolic ability.	Rojas-Tapias et al. [28]
Quorum Sensing & Biofilm Formation	*ffh*, *ftsY*, *yajC*, *ribD*, *wecB*	Contributes to colonization of known pathogens.	Pustelny et al. [29]

## Data Availability

The data generated in this study are publicly available in the NCBI BioSample Database at SAMN30467596.

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
