# Peer review of "Translocation of Oral Microbiota into the Pancreatic Ductal Adenocarcinoma Tumor Microenvironment"

_microorganisms, 2023, doi:10.3390/microorganisms11061466_

Round 1

Reviewer 1 Report

In general, the manuscript has addressed a specific gap in the field.

The study describes the significant assocciation between oral microbiota and microbiota located in the tumor environment.The artcile adds new facts about the role of microorganism in PDAC, especially shows the assocciation between microbiota and PDAC. However, detailed mechanisms did not reveal.

The discussions/conclusions are consistent with the evidence and arguments presented and addressed the main question posed. References are appropriate.

The topic of the article is very interesting. However, presented study has some limitations decreasing the value of article.

1) The cohort study is small.

2) The assessment of with resectable, early-stage PDAC 

3) In the Introduction section, you should describe wider the Veillonells, its characteristic, justification of the choice of this type bacteria

Author Response

Dear esteemed reviewer,

Thank you so much for the insightful reviews of our manuscript. We sincerely appreciate the time taken to read our work and for providing valuable advice on how to improve it.

In general, the manuscript has addressed a specific gap in the field.

The study describes the significant assocciation between oral microbiota and microbiota located in the tumor environment.The artcile adds new facts about the role of microorganism in PDAC, especially shows the assocciation between microbiota and PDAC. However, detailed mechanisms did not reveal.

The discussions/conclusions are consistent with the evidence and arguments presented and addressed the main question posed. References are appropriate.

The topic of the article is very interesting. However, presented study has some limitations decreasing the value of article.

1) The cohort study is small.

Thank you for this comment. While we agree the cohort is small, we believe that the data obtained is impactful for the future of PDAC studies. Our original goal was to assess patient saliva; however, the tumor tissue was included to show similarities to saliva and thus were included in the analysis.

2) The assessment of with resectable, early-stage PDAC 

This is a great point and it has been added into the limitations portion of our manuscript.

3) In the Introduction section, you should describe wider the Veillonells, its characteristic, justification of the choice of this type bacteria

Thank you so much for this comment. We had not included this bacterium in the introduction as this study was not originally designed to focus on any specific bacteria. Veillonella was present in abundance in all of the tumor tissue samples and thus was selected for culture and genomic sequencing. We have since detailed this in the “Culture and sequencing of patient derived Veillonella atypica” portion of the paper and added further information of its characterization into the results for additional clarification.

Reviewer 2 Report

The author compared the relationship between the oral and tumor microbiomes of patients diagnose with PDAC. Using sequencing methods analyzed the salivary and tumor microbiomes. The data showed that a high prevalence and relative abundance of oral bacteria, particullay veillonella and Streptococcus, within the tumor tissue. Veillonella atypica cultured from patient saliva, which have potentially contribute t tumorigenesis. So, they conclude that the taxa found in PDAC tumors may derive from the mouth. These data have potential clinical signficance for diagnose for PDAC.

Major comments:

1. The entire analysis of the microbiome using bioinformatics methods is used to trace the source of tumor microbial strains. Are there any changes that need to be analyzed when oral strains enter the tumor site? It would be more convincing if there were animal level experiments to confirm it.

2. Is it related to the pathological classification of cancer? Is it related to clinical manifestations, complications, and other factors?

Author Response

Dear esteemed reviewer,

Thank you so much for the insightful reviews of our manuscript. We sincerely appreciate the time taken to read our work and the valuable advice on how to improve it.

The author compared the relationship between the oral and tumor microbiomes of patients diagnose with PDAC. Using sequencing methods analyzed the salivary and tumor microbiomes. The data showed that a high prevalence and relative abundance of oral bacteria, particullay veillonella and Streptococcus, within the tumor tissue. Veillonella atypica cultured from patient saliva, which have potentially contribute t tumorigenesis. So, they conclude that the taxa found in PDAC tumors may derive from the mouth. These data have potential clinical signficance for diagnose for PDAC.

Major comments:

  1. The entire analysis of the microbiome using bioinformatics methods is used to trace the source of tumor microbial strains. Are there any changes that need to be analyzed when oral strains enter the tumor site? It would be more convincing if there were animal level experiments to confirm it.

Thank you for this comment. You make an excellent point. This purpose of this study was to explore how the oral microbiome relates to the pancreatic microbiome. Our results show a link between the oral microbiome and pancreatic tumor microbiome. We feel that this data is significant and has the potential to shape future studies, including in murine models. We intend to explore these questions in future studies but feel that it is important to report these exploratory findings to the scientific community.

  1. Is it related to the pathological classification of cancer? Is it related to clinical manifestations, complications, and other factors?

Thank you for this great feedback. The goal of this study was specifically to assess the relationship between the oral microbiome and pancreatic microbiome. This study was exploratory in nature, and the specific taxa highlighted have not previously been described as causal or curative in PDAC specifically. This study gives cause for future studies to include impacts from specific oral bacteria.

Round 2

Reviewer 1 Report

The main limitations of this study are still present.

Reviewer 2 Report

The authors have addressed my previous concerns. No additional comments.